# Mountain Muon Tomography Using a Liquid Scintillator Detector

**Bin Zhang** [1,2] [ID], **Zhe Wang** [1,2,3] [ID] **and Shaomin Chen** [1,2,3,*] [ID]

1 Department of Engineering Physics, Tsinghua University, Beijing 100084, China
2 Center for High Energy Physics, Tsinghua University, Beijing 100084, China
3 Key Laboratory of Particle & Radiation Imaging, Ministry of Education, Tsinghua University, Beijing 100084, China
* Correspondence: chenshaomin@mail.tsinghua.edu.cn

**Abstract:** Muon tomography (MT), based on atmospheric cosmic rays, is a promising technique suitable for nondestructive imaging of the internal structures of mountains. This method uses the measured flux distribution after attenuation, combined with the known muon angular and energy distributions and a 3D satellite map, to perform tomographic imaging of the density distribution inside a probed volume. A muon tomography station (MTS) requires direction-sensitive detectors with a high resolution for optimal tracking of incident cosmic-ray muons. The spherical liquid scintillator detector is one of the best candidates for this application due to its uniform detection efficiency for the whole $4\pi$ solid angle and its excellent ability to distinguish muon signals from the radioactive background via the difference in the energy deposit. This type of detector, with a 1.3 m diameter, was used in the Jinping Neutrino Experiment (JNE). Its angular resolution is 4.9 degrees. Following the application of imaging for structures of Jinping Mountain with JNE published results based on the detector, we apply it to geological prospecting. For mountains below 1 km in height and 2.8 g/cm$^3$ in the reference rock, we demonstrate that this kind of detector can image internal regions with densities of $\leq$2.1 g/cm$^3$ or $\geq$3.5 g/cm$^3$ and hundreds of meters in size.

**Keywords:** cosmic rays; muon tomography; nondestructive imaging; liquid scintillator; geological prospecting





## 1. Introduction

Muon tomography (MT) can nondestructively image a density distribution as deep as several hundred to one thousand meters inside a probed volume, which is an advantage over traditional geological prospecting methods, such as drilling or detecting gamma rays. The data taking time reduces 2–4 times for every 200 m depth reduction.

Extensive research has shown that MT has a wide range of applications, including the imaging of volcanoes [1,2], pyramids [3,4], hidden underground structures [5,6], nuclear reactors [7,8], high-Z materials [9], high-Z and medium-Z materials [10], and bedrock sculpting beneath an alpine glacier [11]. The imaging techniques used in MT are often based on plastic scintillators [12,13], emulsion [14], and gas [15] detectors.

This article focuses on muon geotomography. One pioneering application study [16,17] demonstrated that this is a promising technique for geological prospecting of large-scale objects, such as mountains, to find the internal mineral veins, water layers, air cavities, etc. The methods mentioned above can only simultaneously image mountains within a particular solid angle, which results in a small detection range and, consequently, limits their application in prospecting.

In addition, far too little attention has been paid to liquid scintillators (LS) as a target material. One of the advantages of LS is that the detector volume can be larger, increasing the cosmic-ray counting rate and thus achieving a better imaging resolution under the same data taking time. Additionally, the large target mass enables it to make the best use of

sufficient energy deposit information in separating the muon signal from the background (dominantly natural radioactive beta decay). Moreover, the detector's structure can easily be constructed spherically, which enables it to have uniform detection efficiency and angular resolution for the $4\pi$ solid angle. Therefore, the muon flux and direction measurements can directly reflect the density distribution inside a mountain without efficiency correction and solve the problem of a small detection range.

A one-ton spherical liquid scintillator detector with a 1.3 m diameter was used in the Jinping Neutrino Experiment (JNE). In this paper, we study its application in geological prospecting. Such a small size makes it movable, enabling changes to the observation locations flexibly. Its angular resolution is 4.9 degrees.

In this article, we report on an MT study using the JNE's published results with this detector from the China Jinping Underground Laboratory (CJPL) [18], performing the imaging of the structure of Jinping Mountain. In addition, an extension of this study, through a simulation based on Geant4 [19,20] to observe the density variation inside a mountain with a height (defined as the vertical distance from its bottom to its top) of less than 1 km, shows that this type of detector has broad application prospects within geological prospecting.

## 2. Materials and Methods

### 2.1. Cosmic-Ray Muon and Its Interaction with Matter

Primary cosmic rays (dominantly protons) interact with matter in the upper atmosphere, creating enormous amounts of short-lived $\pi$ and K mesons, which decay to produce cosmic-ray muons.

#### 2.1.1. Energy and Angular Distribution at Sea Level

Gaisser [21] introduced a formula to describe the energy and angular distributions of muons flying to sea level. Considering muon decay and the curvature of the Earth, Guan et al. [22] proposed a modified Gaisser's formula that expanded its applicability to low energies and all zenith angles:

$$\frac{dN_\mu}{dE_\mu d\Omega} = \frac{0.14}{\text{cm}^2 \text{ s sr GeV}} \left[ E_\mu + \frac{3.64 \text{ GeV}}{(\cos\theta^\star)^{1.29}} \right]^{-2.7} \times \left[ \frac{1}{1 + \frac{1.1E_\mu\cos\theta^\star}{115 \text{ GeV}}} + \frac{0.054}{1 + \frac{1.1E_\mu\cos\theta^\star}{850 \text{ GeV}}} \right], \quad (1)$$

where $N_\mu$ is the muon flux, $E_\mu$ is the kinetic energy (hereinafter referred to as energy), $\Omega$ is the solid angle, and $\cos\theta^\star$ is defined as follows:

$$\cos\theta^\star = \sqrt{\frac{(\cos\theta)^2 + 0.102573^2 - 0.068287(\cos\theta)^{0.958633} + 0.0407253(\cos\theta)^{0.817285}}{1 + 0.102573^2 - 0.068287 + 0.0407253}}, \quad (2)$$

where $\theta$ is the zenith angle. Figure 1 is the energy projection of Formula (2), and the minimum energy is taken as 1 GeV for simplicity without loss of generality. In the figure, the energy range of more than 100 GeV is what we studied, and its quantity accounts for 0.7%. Muons of this energy can easily pass through a thickness of matter of 150 m $\times$ 2.8 g/cm$^3$ or more and are the primary sources of the signals in this study. We use an estimated value of 2.8 g/cm$^3$ as the reference density for mountain rock throughout this article. We do not consider the contributions of muons with energies of less than 100 GeV because the superficial part of the mountain too quickly blocks them from reaching the detector. The interest of this study is to image the deeper part.

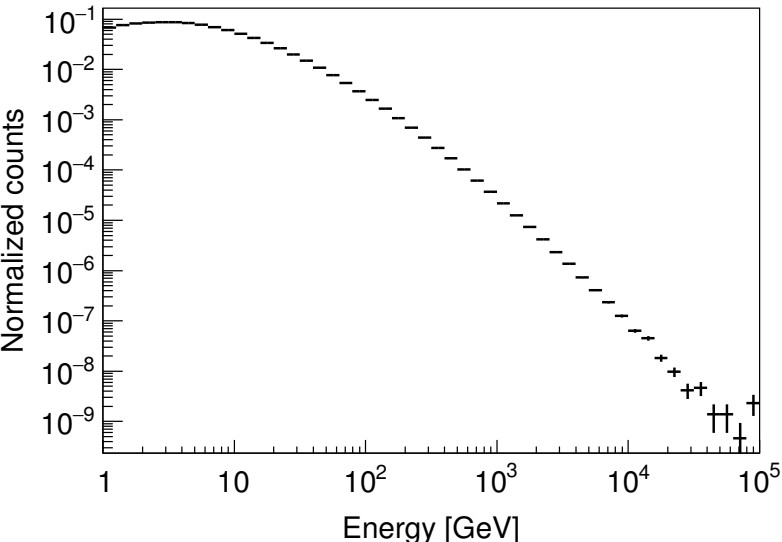

**Figure 1.** Distribution of sea-level muon kinetic energy.

As pointed out by Cecchini et al. [23], the angular and energy distributions vary little with the latitude and altitude of the muons that have small zenith angles and energies higher than 40 GeV, which is below the lower limit of 100 GeV of the studied energy range. In addition, for a large zenith angle, the muon is close to flying in the horizontal direction. Its trajectory length in the mountain is so long that the counting rate is limited, resulting in a minor contribution to the imaging. Furthermore, for mountain heights below 1 km, the altitude of the mountain top is typically a few kilometers, far less than the 30 km altitude where the muons are produced. Therefore, in this study, we assume that the muons, before entering into mountains, still follow the distribution represented by Formula (2), even though they are above sea level.

In this study, we also ignore the possible deviation between the actual differential flux and the modified Gaisser's formula.

2.1.2. Flux Attenuation in Matter

The energy deposit caused by the interaction between the muon and the mountain matter, such as rock, depends on the opacity [24]:

$$\mathcal{O} \equiv \int_L \rho(\xi) \mathrm{d}\xi, \tag{3}$$

where $\rho$ is the matter density, and $\xi$ is the coordinate along the trajectory $L$ that the muon travels through the mountain. Muons with energies lower than the energy deposit are blocked by the mountain from reaching the detector, causing a flux attenuation. For a particular mountain, the opacity varies with the direction viewed from the detector such that the flux attenuation varies with the direction. We obtain the mountain opacity distribution by measuring the flux and direction, i.e., we conduct a mountain MT.

Figure 2 shows the distribution of the opacity $\mathcal{O}$ of the matter with which sea-level muons interact until they either stop or decay. It can be used to estimate the event rate for a given rock overburden above the detector. We find that the curve in the figure can be approximated to a straight line when $\mathcal{O} \geq 2.8 \, \mathrm{g} \cdot \mathrm{km/cm^3}$, which indicates that the flux attenuates exponentially as the opacity increases. The slope of the curve is larger when $\mathcal{O} \leq 2.8 \, \mathrm{g} \cdot \mathrm{km/cm^3}$, which indicates that the attenuation is faster than the exponential.

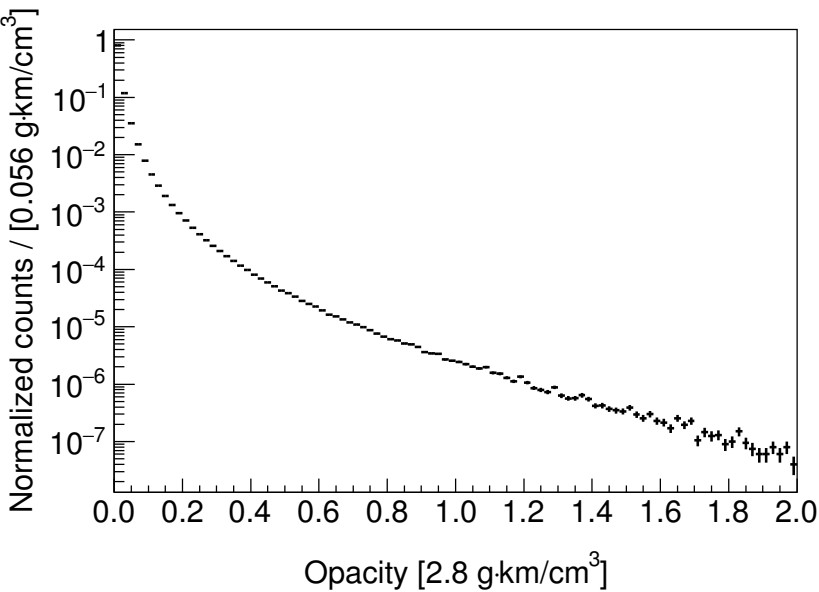

**Figure 2.** Distribution of opacity of the matter with which sea-level muons interact until stopping or decaying.

It is worth noting that the lower limit of the studied energy range is 100 GeV, which is far greater than a muon's mass of about 0.1 GeV, making the muon an extremely relativistic particle, and the energy deposit in a unit opacity is independent of the energy.

### 2.2. Muon Tomography Using a Liquid Scintillator Detector

Liquid scintillator detectors are commonly applied to neutrino experiments in underground laboratories worldwide. Since this type of detector has demonstrated excellent performance in measuring cosmic-ray muons, it can also provide muon tomography.

### 2.2.1. Detector and Dataset

A spherical liquid scintillator detector [25] with a target mass of one ton was built and operated in the JNE, and its structure has been adopted by the simulations of this study. Figure 3a shows its external structure. The outermost structure is a white cylindrical steel tank of 2 m in height and 2 m in diameter. The internal structure is shown in Figure 3b. Thirty inward-facing 8-inch photomultiplier tubes (PMTs) mounted on the blacklight shading sphere detect the emitted scintillation light to measure the particle energy deposit. Inside the light shading sphere is a transparent acrylic sphere with a diameter of 1.3 m that contains one-ton liquid scintillator. The remaining space between the acrylic sphere and the steel tank is filled with water as a buffer to block the radioactive background. The electronics system uses one logic trigger module of CAEN V1495 and four FlashADC boards of CAEN V1751. The FlashADC boards can sample the electrical signals of the PMTs as waveforms for subsequent offline analysis at a sampling rate of 1 GHz. The time information extracted from the waveform is used for the subsequent muon direction reconstruction.

The detector has been placed in the CJPL with a 2400 m rock overburden. Using the detector, the JNE's collaboration [26,27] detected 343 cosmic-ray muon events over 820.3 days and measured their direction distribution based on a direction reconstruction method. This method and the measured distribution will be introduced in Sections 2.3 and 3.1, respectively. The background of the muon signal is dominantly natural radioactivity with an energy deposit of less than 10 MeV and is much lower than the minimum 100 MeV required by the cosmic-ray muon event selection criteria. Consequently, this background can be suppressed to a negligible level.

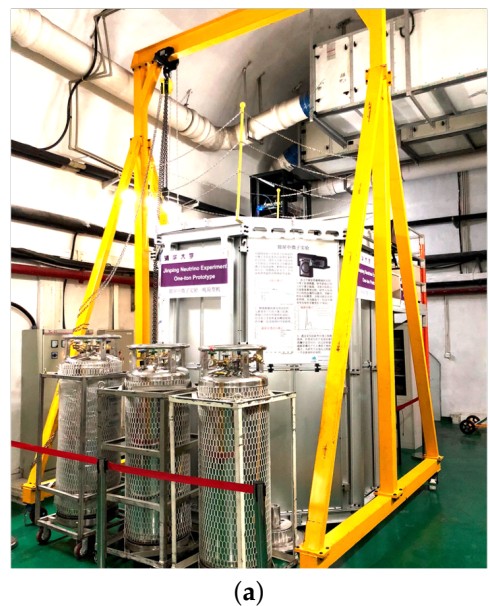
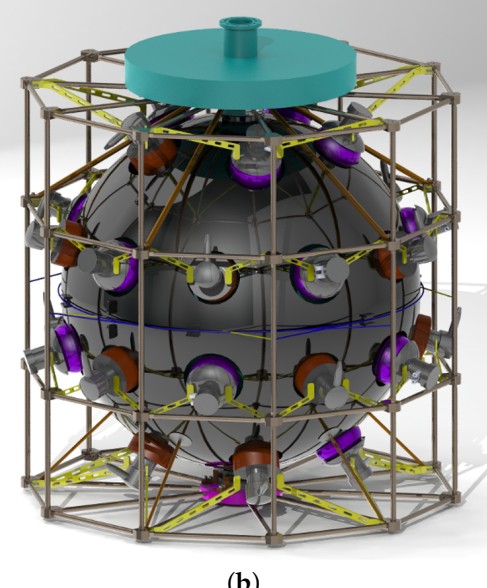

(**a**)　　　　　　　　　　　　　　　　　　　　　(**b**)

**Figure 3.** Views of the one-ton spherical liquid scintillator detector: (**a**) external structure and (**b**) internal structure.

### 2.2.2. Event Rate Calculation

In simulations, the setup of the cosmic-ray event rate in the detector is crucial for the acquisition of the data taking time required to achieve a specific imaging resolution. For the calculation of the event rate in a detector placed somewhere inside or outside the mountain, the first parameter that we need to know is $R_0$, which is defined as the event rate of the detector when there is no geological object above the plane in which it is located. $R_0$ depends on the detector's parameters, including the geometry, size, target mass, muon detection efficiency, and so on. The second parameter is the survival probability $P$, which is defined as the ratio of cosmic-ray muon fluxes before and after attenuation in the mountain. We can obtain $P$ via simulations based on a satellite image of the mountain terrain, the reference rock density, and the detector's position. The product of $R_0$ and $P$ gives the calculated event rate. The event number, namely the product of the event rate and the data taking time, affects the imaging resolution. In addition, $R_0$ is essential for the acquisition of the reference radar image, which is defined in Section 2.4.

There are two methods to calculate the $R_0$ of the one-ton detector:

1.　As mentioned in Section 2.2.1, the measured event rate is

$$R_{\text{CJPL}} = (48.4 \pm 2.6_{\text{stat.}}) \times 10^{-7} \text{ Hz} \tag{4}$$

at the CJPL inside Jinping Mountain. Based on a satellite image of the Jinping Mountain terrain, obtained from the NASA SRTM3 dataset [28], and the reference density, we obtain the survival probability $P_{\text{CJPL}} = 2.1 \times 10^{-8}$. Furthermore, we have:

$$R_0 = R_{\text{CJPL}}/P_{\text{CJPL}} = (230 \pm 12_{\text{stat.}}) \text{ Hz}. \tag{5}$$

2.　The muon flux at sea level is known to be about $1 \text{ cm}^{-2}\text{min}^{-1}$. We assume that the sensitive area of the detector is $\frac{1}{4}\pi(1.3 \text{ m})^2$ and the detection efficiency is 100%. Then, $R_0$ is about 220 Hz.

The conclusions of the above two methods are consistent. Therefore, we use $R_0 = 230$ Hz in the following simulations. In actual mountain imaging, we can place the detector on the open ground and measure a more accurate $R_0$.

### 2.3. Direction Reconstruction

One of the foundations of the MT that uses this kind of detector is reconstructing muon direction. We use a template reconstruction method based on the time measurement for each PMT, as mentioned in Section 2.2.1. It relies on the fact that when a muon passes through the detector, its direction and entry point determine the emission times and positions of the resultant scintillation photons along the trajectory, thereby determining the photons' arrival times on the PMTs. Therefore, we can extract the direction from the arrival times.

### 2.3.1. Reconstruction Templates' Creation

We use simulations to create templates for reconstructing cosmic-ray muon events. The parameters of a template include the true values of the direction and the entry point on the spherical detector shell, as well as the time information on the PMTs, which is recorded as it generates a response in the detector. The number of templates is about 500 k, which we expect to fully cover all of the possibilities for the muon passing through the detector. For the $n$-th template, we randomly sample the direction $\mathbf{P}_n = (\cos\theta_n, \phi_n)$ and the entry point $(\cos\alpha_n, \beta_n)$. The $\theta_n$ and $\alpha_n$ are zenith angles, while the $\phi_n$ and $\beta_n$ are azimuth angles.

$\mathbf{P}_n$ and $(\cos\alpha_n, \beta_n)$ serve as inputs to a detector response simulation (using a Geant4-based software package). From the output, we can obtain the time $\tau_n^i$ at which the scintillation light, emitted by interactions between the incident muon and the LS, arrives at the $i$-th PMT. Based on $\tau_n^i$ ($i = 1, 2, \ldots, N_{\text{PMT}}$), we have an $N_{\text{PMT}}$-dimensional zero-centered arrival time vector $\mathbf{T}_n$, where $N_{\text{PMT}}$ is the number of the PMTs. $\mathbf{P}_n$, $(\cos\alpha_n, \beta_n)$, and $\mathbf{T}_n$ are calculated as follows:

- $\mathbf{P}_n = (\cos\theta_n, \phi_n)$: The $\cos\theta_n$ and $\phi_n$ values are sampled from uniform distributions between $-1$ and $1$ and between $0$ and $2\pi$, respectively;
- $(\cos\alpha_n, \beta_n)$: The entry point is sampled uniformly from the detector hemisphere facing the muon direction $\mathbf{P}_n$;
- $\mathbf{T}_n$: $\tau_n^i$ ($i = 1, 2, \ldots, N_{\text{PMT}}$) contains a global time that records when the muon arrives at the detector. The global time does not contain the direction information, and we remove it via centralization to zero:

$$t_n^j \equiv \tau_n^j - \frac{1}{N_{\text{PMT}}} \sum_{i=1}^{N_{\text{PMT}}} \tau_n^i, \tag{6}$$

and we can define:

$$\mathbf{T}_n \equiv \left( t_n^1, \cdots, t_n^{N_{\text{PMT}}} \right). \tag{7}$$

### 2.3.2. Process of Reconstruction

For a detected event that is ready to be reconstructed, we fill the time information into the zero-centered arrival time vector $\mathbf{T}_{\text{rec}} \equiv \left( t_{\text{rec}}^1, \cdots, t_{\text{rec}}^{N_{\text{PMT}}} \right)$ in exactly the same way as the template event. Then, we define the distance between this event and the $n$-th template event as $d_n$:

$$d_n \equiv \| \mathbf{T}_{\text{rec}} - \mathbf{T}_n \| = \sqrt{\sum_{i=1}^{N_{\text{PMT}}} \left( t_{\text{rec}}^i - t_n^i \right)^2}. \tag{8}$$

We assume that the smaller the distance $d_n$, the more similar the $n$-th template will be to the event. The reconstructed direction is the weighted sum of the momentum directions of the $k$ templates with the smallest distance $d_{n_m}$:

$$\mathbf{P}_{\text{rec}} = \frac{\sum_{m=1}^{k} W_{n_m} \mathbf{P}_{n_m}}{\sum_{m=1}^{k} W_{n_m}}, \tag{9}$$

where the weight is:

$$W_{n_m} \equiv 1/d_{n_m}, \tag{10}$$

and $n_m$ is the index of the template with the $m$-th smallest distance. Here, $k$ is determined to be the value that minimizes the angular resolution, and the resolution is defined as the mean of the included angle $\Delta\Theta$ between the reconstructed direction and the true direction.

### 2.3.3. Performance of the Method

We take $N_{\text{PMT}}$ to be 30, which is the same as the number of PMTs in the one-ton detector. In order to test the performance, we generate a test dataset in the same way as the template dataset, reconstruct their directions, and calculate the included angle $\Delta\Theta$ between the true and the reconstructed directions. We find the mean of $\Delta\Theta$ reaches its minimum when $k = 18$, which this study adopts. According to whether $\Delta\Theta > 90°$, we divide the dataset into two parts: unsuccessfully or successfully reconstructed events. The former accounts for 0.1% of the events, and the mean $\Delta\Theta$ of the latter is 4.9 degrees, which is the angular resolution of the detector.

### 2.4. *Principle and Analysis Process of the Muon Tomography*

We obtain the ratio radar image by dividing the measured and referenced event number radar images. The ratio radar image shows the locations of the density variation regions inside the mountain. The above radar images are two-dimensional (2D); the radial and angular dimensions are $\cos\theta_{\text{zen}}$ and $\phi_{\text{azi}}$, respectively. The $\theta_{\text{zen}}$ and $\phi_{\text{azi}}$ are, correspondingly, the zenith angle and the azimuth angle of the muon direction in a spherical coordinate system with the detector as the center and a vertically upward direction as the positive Z-axis. The event number radar image or the ratio radar image indicates that the bin content is the event number or the ratio of the event numbers.

Figure 4 shows the analysis flow chart. The radar images can be obtained in the following way: We place the detector in a position and then take cosmic-ray muon data for a given time to obtain the measured event number radar image. Based on the satellite image of the mountain terrain, the reference density obtained by measuring the mountain rock, the $R_0$, the data taking time, Formula (2), and the direction reconstruction method, we can obtain the referenced event number radar image via simulations. After normalization by the data taking time, dividing the measured image by the reference one, yields the ratio radar image, which displays the 2D locations of the density variation regions. In those locations, we require the quotient to deviate from one with a statistical significance of at least three standard deviations. The detector can be placed at two or more measurement positions to obtain a 3D image of the density variation regions by combining the 2D images if one is interested.

In Section 3, we will present the imaging of Jinping Mountain based on the JNE's published results, as mentioned in Section 2.2.1, and the simulated data. For the imaging with the JNE's published results, the total number of events is 343, too few to give a statistically significant ratio radar image, so we give an event number radar image. For the imaging with simulated data, we will give the ratio radar images of two scenarios in which the detector is placed inside and outside the mountain.

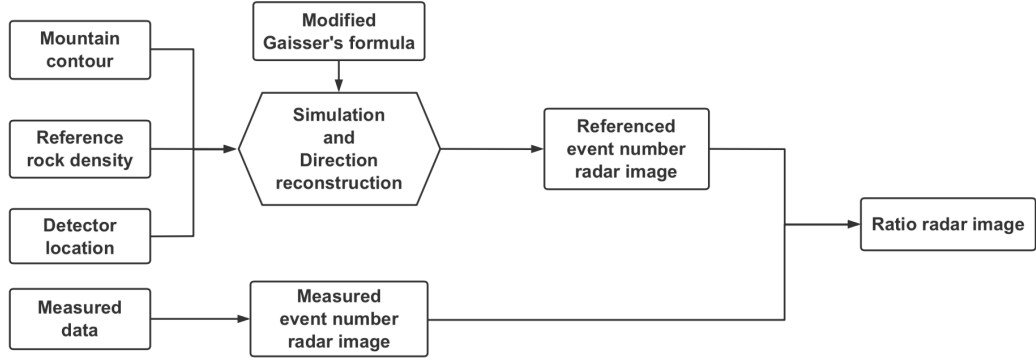

**Figure 4.** Analysis flow chart of the mountain MT.

## 3. Results and Discussion

In this section, we obtain a rough internal imaging of Jinping Mountain using the one-ton detector. Extending the study, we find that the liquid scintillator detector can provide a good muon tomography, which can be applied to geological prospecting.

### 3.1. Structural Imaging of the Mountain Using the JNE's Published Results

Figure 5a shows the satellite image of the Jinping Mountain terrain, and Figure 5b shows the event number radar image based on the JNE's published results, as mentioned in Section 2.2.1. There are event number excesses in four main directions (two in the south and two in the north) in the radar image, which match the ravines seen in the satellite image.

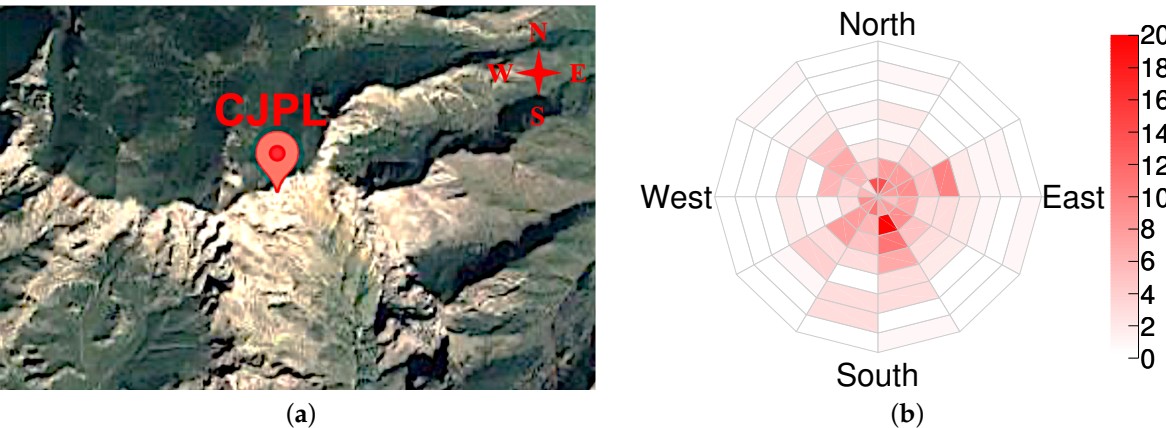

(**a**)　　　　　　　　　　　(**b**)

**Figure 5.** The satellite image of the Jinping Mountain terrain (**a**) was extracted from Google Maps. In the measured event number radar image (**b**), the color represents the event number in each bin. The radial dimension is the $\cos\theta_{zen}$ of the muon direction; the innermost point and the outermost circle correspond to $\cos\theta_{zen} = 1$ (vertical upward direction) and $\cos\theta_{zen} = 0.2$, respectively. There are event number excesses in four main directions (two in the south and two in the north) in the radar image, which match the ravines seen in the satellite image.

### 3.2. Internal Imaging of the Mountain Using Simulated Data

The statistics of the JNE's published results are too low to give a statistically significant imaging of the mountain due to its deep rock overburden (2.4 km). Therefore, to extend the study from the existing result at the CJPL, we vertically up-shift the detector to a location with a 1 km rock overburden and set the overall density of the mountain to the reference one to study its internal imaging through simulations. We expect a more than 100-fold increase in the event rate, according to Figure 2. We find the ratio radar image can statistically significantly show the locations of the density variation regions as the data taking time reaches six months for the following geometry and density settings of the regions.

The imaging quality depends on the parameters of the density variation regions, including the density, the geometric size, the thickness of the rock overburden, the distance from the detector, and the zenith angle of the regions viewed from the detector. For simplicity, we only study the density effect.

### 3.2.1. Scenario 1: The Detector Is Placed inside the Mountain

Figure 6 shows the shapes and locations of the density variation regions. The mountain height is set to 1 km, and the detector is placed at the center of the bottom surface of the mountain. In actual imaging, this is suitable for mountains with traffic tunnels, in which the detector can be placed. The eight density variation regions are set to be located within the same cylinder, which has a height of 0.2 km, a diameter of 2 km, and a center located at 0.5 km above the detector and is completely wrapped in the mountain. The setting of multiple small regions helps to study whether the position of each small region can be displayed in the radar image to better reflect the imaging capability of this MT.

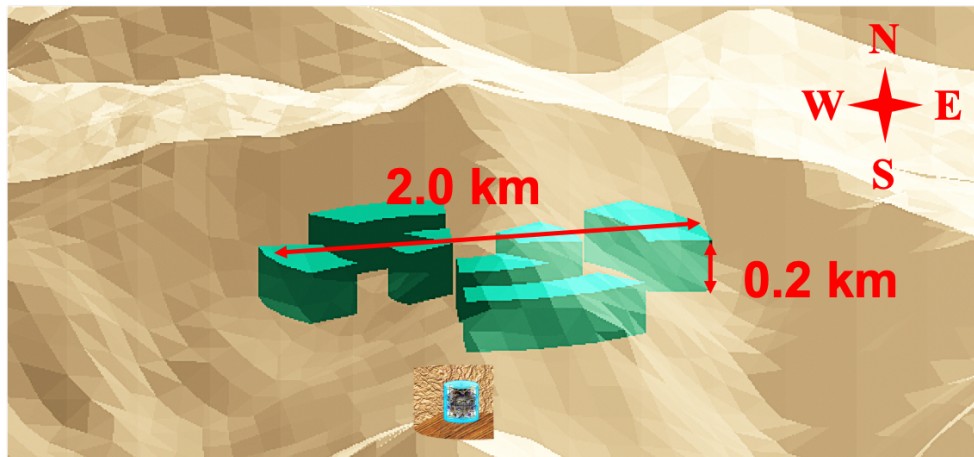

**Figure 6.** The mountain (grey) with artificially embedded density variation regions (cyan) and their shapes and locations. The height and diameter of the detector are enlarged 100 times for display convenience. The regions are located in the same cylinder, the center of which is 500 m above the detector.

Through the method described in Section 2.4, we obtain the ratio radar image of the density variation, as shown in Figure 7.

For the six-month data taking time and at least a 50% ($\leq 1.4$ or $\geq 4.2$ g/cm$^3$) density variation from the reference one, we find that the colored areas in the radar image successfully show the locations of the density variation regions and that some small regions can be identified, indicating that the MT can identify the geometric structure on a hundred-meter scale. The size of the density variation regions is defined as the average muon trajectory length when the regions' density is set to the reference density. Calculations show that the size is 0.16 km. We propose to determine whether the colored areas are caused by density variations by whether the colored bins are clustered in the radar image. This criterion can distinguish the false signals caused by statistical fluctuations where the bins are randomly distributed in the radar image. The greater the density deviation, the greater the number of colored bins, the darker the color, and the better the imaging performance. The extraction of the numerical value of the density from the color, i.e., the ratio *r* of the event numbers, needs to be studied further. However, we can ascertain whether the density is larger or smaller than the reference one, as this corresponds to the ratio *r* being smaller or larger than one. The locations of the density variation regions displayed on the radar image can guide subsequent prospecting, such as drilling.

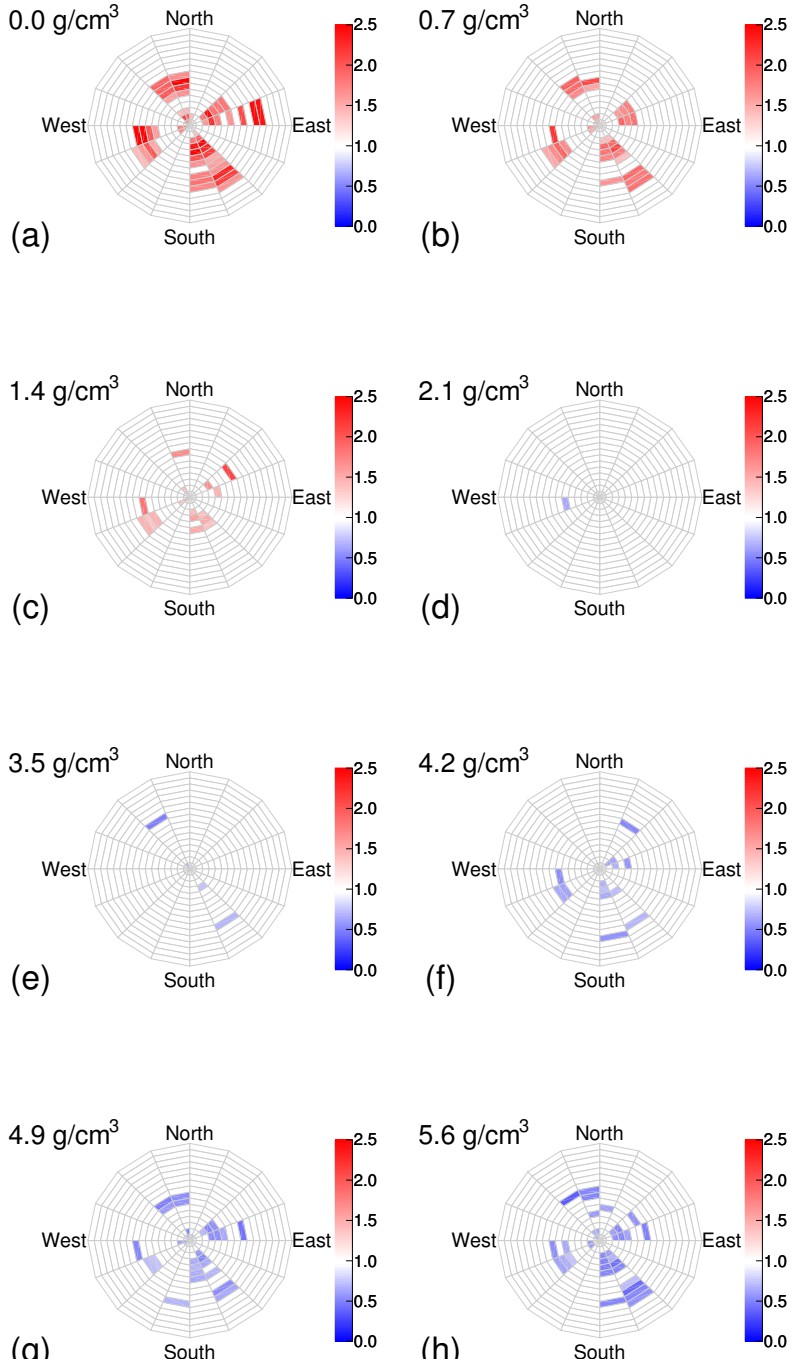

**Figure 7.** The ratio radar images of the density variation regions, assuming a six-month data acquisition with the detector placed inside the mountain. The radial dimension is the $\cos\theta_{zen}$ of the muon direction; the innermost point and the outermost circle correspond to $\cos\theta_{zen} = 1$ (vertical upward direction) and $\cos\theta_{zen} = 0.2$, respectively. We do not draw the regions with a $\cos\theta_{zen} < 0.2$, in which the MT is not sensitive due to the low event rate. The densities of (**a**–**h**) are 0, 0.7, 1.4, 2.1, 3.5, 4.2, 4.9, and 5.6 g/cm$^3$, respectively. The color shows the ratio $r$ of the number of muons passing through the mountain with the density variation regions to the number of muons passing through the reference mountain. The division is weighted according to the data acquisition time. We set the $r$ of the bins with a statistical significance $< 3\sigma$ to 1, which means no difference from the reference one, when plotting. The ratio uncertainty $r_{unc}$ of the colored bins satisfies $r_{unc} \leq |r - 1|/3$.

The total event numbers are 19.5 k, 15.5 k, and 13.1 k when the regions' densities are set to 0, 2.8 (reference), and 5.6 g/cm$^3$, respectively. The difference between the event number and the reference one can be used to test the ability of the MT to distinguish the density variation regions. The event rate ratio of the 1 km and 2.4 km overburdens is

$$\lambda = \frac{15500}{0.5 \times 365} \Big/ \frac{264}{645.2} \approx 208, \tag{11}$$

which is consistent with the estimation, according to Figure 2.

### 3.2.2. Scenario 2: The Detector Is Placed outside the Mountain

We place the detector on the surface of a ravine of the mountain to reduce the flux attenuation due to the mountain, thereby increasing the event rate. The mountain height is set to 1 km. Figure 8 shows the shape and location of the density variation region. Compared with Scenario 1, in which the detector is placed inside the mountain, the imaging here is more difficult, mainly for the following two reasons. Firstly, the distance from the detector to the center of the density variation region increases such that the muon flux from the region decreases. Secondly, the zenith angle of the region viewed by the detector increases and further reduces the flux due to the angular dependence of cosmic-ray muons: the greater the zenith angle, the lower the flux. Therefore, we up-shift the region's position, thereby reducing the zenith angle; however, the location is closer to the mountain top. Hence, the horizontal size of the region must be reduced, and, therefore, we increase its height to 0.4 km to compensate. To make the region completely wrapped by the mountain, we set its shape as a circular truncated cone. To balance the contradiction between the distance and the zenith angle, we place the detector in a position with a horizontal distance of 1.5 km and a vertical distance of 0.6 km from the density variation region's center, and the region is located in the northeast of the detector.

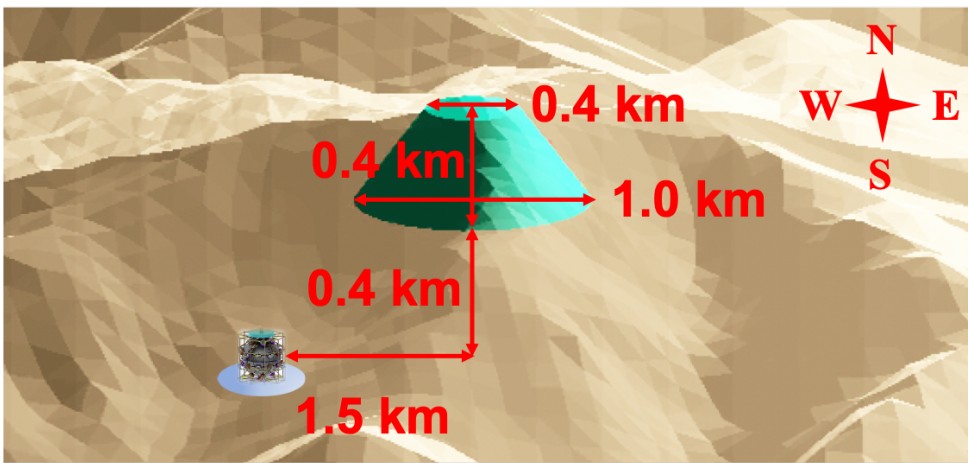

**Figure 8.** The mountain (grey) with an artificially embedded density variation region (cyan) and its shape and location. The height and diameter of the detector are enlarged 100 times for display convenience. The horizontal distance from the detector to the center of the region is 1.5 km.

Compared with the case in which the detector is placed inside the mountain, due to the lack of mountain shielding, a large number of muons from the sky can reach the detector directly without attenuation. Such muons have a high event rate and do not carry information about the internal structure of the mountain. Their reconstructed directions will deviate from the actual path to the mountain direction by a considerable proportion, which increases the background rate. Installing an anti-coincidence detector (for example, a plastic scintillator) facing the sky will contribute to solving this problem.

For the inner liquid scintillator detector, the time window of about 20 μs after the muon event is not available due to electronic system restoration and other reasons. When

there is a signal in the anti-coincidence detector, we open an anti-coincidence time window, set its width to 20 µs, and mark the signal within it. Such signals, while not useful for direct mountain imaging, are useful for event rate calibration, thus indirectly contributing to the technique. The fraction of the dead time (i.e., the period that a detector stops functioning) in the detection is

$$\eta_{\mathrm{dead}} = 230\,\mathrm{Hz} \times 20\,\mathrm{µs} \times 100\% \approx 0.5\%, \tag{12}$$

which is so small that we can ignore its contributions to this study. For the sake of simplicity, we assume that the muons coming directly from the sky are within the detection range of the anti-coincidence detector, so their contribution is not considered. We ignore the non-uniformity of the detection efficiency caused by the anti-coincidence detector because it has little effect on the mountain center (in which we are more interested).

Through the method described in Section 2.4, we obtain the ratio radar image of the density variation, as shown in Figure 9.

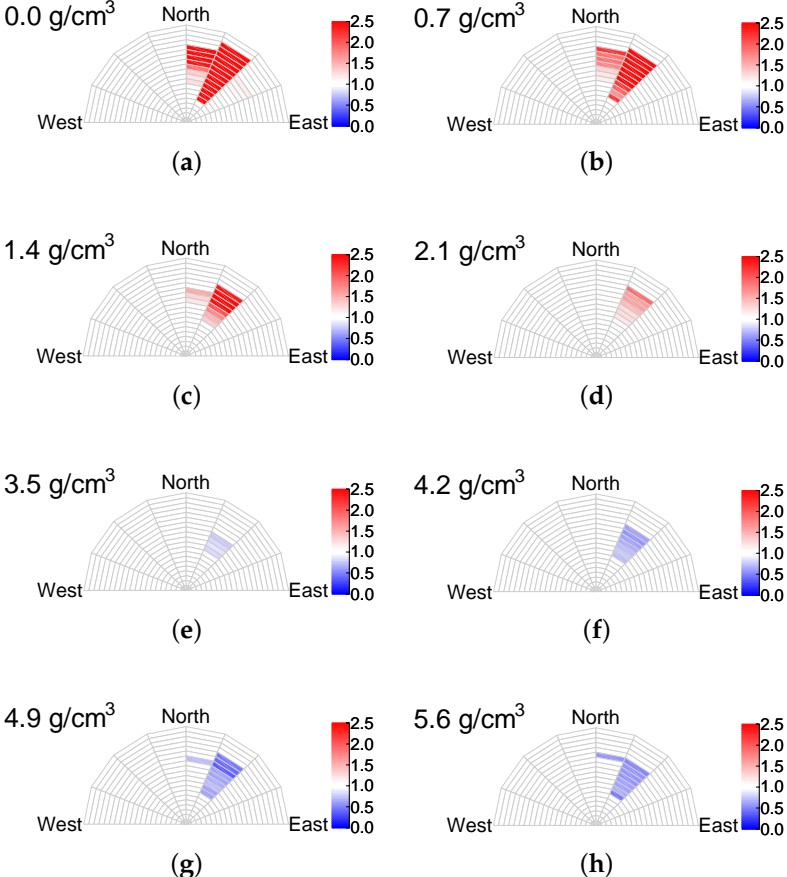

**Figure 9.** The ratio radar images of the density variation region, assuming a six-month data acquisition period with the detector placed outside the mountain. The radial dimension is the $\cos\theta_{\mathrm{zen}}$; the innermost point and the outermost circle correspond to the vertical ($\cos\theta_{\mathrm{zen}} = 1$) and horizontal ($\cos\theta_{\mathrm{zen}} = 0$) direction of muon momentum, respectively. The southern part of each image is not drawn because it is within the detection range of the anti-coincidence detector. The densities of (**a**–**h**) are 0, 0.7, 1.4, 2.1, 3.5, 4.2, 4.9, and 5.6 g/cm$^3$, respectively. The color shows the ratio $r$ of the number of muons passing through the mountain within the density variation regions to the number of muons passing through the reference mountain. The division is weighted according to the data acquisition time. We set the $r$ of the bins with the statistical significance $< 3\sigma$ to 1, which means there is no difference from the reference one, when plotting. The ratio uncertainty $r_{\mathrm{unc}}$ of the colored bins satisfies $r_{\mathrm{unc}} \leq |r - 1|/3$.

For a six-month data taking time and at least a 25% ($\leq$2.1 or $\geq$3.5 g/cm$^3$) density variation from the reference one, we find that the colored areas in the radar image successfully show the locations of the density variation region. The southern part of each image is not drawn because it is within the detection range of the anti-coincidence detector. The size of the region is 0.30 km, as defined in Section 3.2.1. The greater the density deviation, the greater the number of colored bins, the darker the color, and the better the imaging performance.

The total event numbers are 81.9 k, 60.5 k, and 57.1 k for the region densities set to 0, 2.8 (reference), and 5.6 g/cm$^3$, respectively. The difference between the event number and the reference one can be used to measure the ability of the MT to distinguish the density variation regions.

## 4. Conclusions

A one-ton liquid scintillator detector is initially used for a prototype of a neutrino experiment, which can also serve as a mountain MT based on muon absorption. Using the JNE's published results measured under 2.4 km from the top of Jinping Mountain, we demonstrate this kind of detector's capacity in examining the mountain structures from a 2D event number radar image. We extend the imaging study to mountains below 1 km in height and with a 2.8 g/cm$^3$ reference rock density. Assuming a 6-month data acquisition period, we conclude that when the detector is placed inside the mountain, the MT can spot the regions with a size of 0.16 km and a density variation of more than 50% ($\leq$1.4 or $\geq$4.2 g/cm$^3$) from the reference one and can image their geometric structure. If placing the detector on the mountainside, the identified density variation threshold reduces to 25% ($\leq$2.1 or $\geq$3.5 g/cm$^3$) when the size is set to 0.30 km.

Our study shows that typical density variations compared to the reference value, such as air (0 g/cm$^3$), water (1 g/cm$^3$), and chalcocite veins (5.5 g/cm$^3$), are all within the sensitive detection range of this kind of MT. We, therefore, anticipate that it will have broad application prospects in geological prospecting.

**Author Contributions:** Investigation, B.Z.; Supervision, S.C.; Validation, Z.W. All authors have read and agreed to the published version of the manuscript.

**Funding:** This work was supported in part by the National Natural Science Foundation of China (12127808, 12141503).

**Institutional Review Board Statement:** Not applicable.

**Informed Consent Statement:** Not applicable.

**Data Availability Statement:** Not applicable.

**Acknowledgments:** We are grateful to Linyan Wan, Ziyi Guo, and Lei Guo for the development of the simulation software package. We appreciate Lin Zhao, Yiyang Wu, Aiqiang Zhang, Jun Weng, Wentai Luo, and Benda Xu for their advice. This work was supported in part by the Key Laboratory of Particle and Radiation Imaging (Tsinghua University).

**Conflicts of Interest:** The authors declare no conflict of interest.

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
