# Peer review of "Mountain Muon Tomography Using a Liquid Scintillator Detector"

_applsci, doi:10.3390/app122110975_

Round 1

Reviewer 1 Report

This manuscript studies the muon tomography potential on geological prospecting with a liquid scintillator detector used in Jinping Neutrino Experiment. The detector, method, data and synthetic tests are described. This manuscript can be accepted with some moderate revisions.

 (1) Since the mountain has very large thickness, the muon flux is very small. The number of observed muon events are also rather small. So please provide the uncertainty for the ratio radar images in the synthetic tests.

(2) The artificially embedded density variation regions in the mountain are used in the synthetic tests. Are there any consideration for building such anomalies inside the mountain?

Author Response

(1)Since the mountain has very large thickness, the muon flux is very small. The number of observed muon events are also rather small. So please provide the uncertainty for the ratio radar images in the synthetic tests.

Agree. We add one sentence into the captions of Figure 7 & 8: “The ratio uncertainty $r_{\rm unc}$ of colored bins satisfies: $r_{\rm unc}\leq|r-1|/3$.”

(2)The artificially embedded density variation regions in the mountain are used in the synthetic tests. Are there any consideration for building such anomalies inside the mountain?

Agree. For further application study, we do have this consideration.

Reviewer 2 Report

Dear authors,

I found your manuscript really interesting, mostly for the type of detector you used in your research. It's not mandatory, but I would recommend making a comparison between it and the detector that are usually used in muographic applications.

Now I proceed with a point-to-point review of your manuscript:

- first of all, please check English language, both to better express the concepts and for some spelling errors.

-  line 8 -- even if the detector has a 4 pi solid angle view, the muon flux cannot come from below. 

- lines 17 -> 21 -- please spend some word about data acquisition time in relation to the thickness crossed by muons.

- line 38 -- 4 pi -> 2 pi

- line 73 -- I would say that although the muon flux varies with latitude and altitude, this is negligible for muographic purposes.

- line 77 -- meager -> limited?

- lines below 298 -- can you express the dead time, more than the proportion?

I thank you in advance.

Author Response

1. It's not mandatory, but I would recommend making a comparison between it and the detector that are usually used in muographic applications.

We apologize we didn’t emphasize this article aims to the application of Muon Geotomography, imaging large-scale objects. Other muon tomography applications are only used in imaging of fine structure of small objects. What we can compare is the technique using a gas tracking detector, which unfortunately we didn’t cite in our original manuscript. Comparing to our detector, it cannot have a 4 pi solid angle coverage of detection.

2. first of all, please check English language, both to better express the concepts and for some spelling errors.

Agree.

3. line 8  & 38-- even if the detector has a 4pi solid angle view, the muon flux cannot come from below.  & line 38 -- 4 pi -> 2 pi

We insist using 4 pi solid angle since the other 2 pi angle is used as a check.

4. lines 17 -> 21 -- please spend some word about data acquisition time in relation to the thickness crossed by muons.

Agree. We add one sentence at the end of Line 21: “The data taking time will reduce 2-4 times for every 200~m depth reduction.”

5. I would say that although the muon flux varies with latitude and altitude, this is negligible for muographic purposes.

You are right, we want to illustrate this fact in this line.

6. line 77 -- meager -> limited?

Fixed.

7. lines below 298 -- can you express the dead time, more than the proportion?

Ok, we revise the sentence to clarify what the dead time means. “The proportion of dead time in the measurement is” is modified by “The fraction of the dead time (namely the period that a detector stops functioning) in the detection is”.

Reviewer 3 Report

- perhaps to add reference for the rock density 2.8 g/cm3

- style: references 1 and 13 are the same

- line 115: 0.645 m is the diameter or the radius of the sphere?

Author Response

1. perhaps to add reference for the rock density 2.8 g/cm3

This is from a measurement of marbles, unfortunately the lab doesn’t have a reference for us to cite.

2. style: references 1 and 13 are the same

Fixed.

3. line 115: 0.645 m is the diameter or the radius of the sphere?

Fixed, the right diameter is 1.3 m.

Reviewer 4 Report

This research is a study to apply a liquid scintillator to muon tomography, and the reviewer recognized that this study attempts to examine a new possibilities of muon tomography.

On the other hand, the reviewers judged that this manuscript lacked scientific significance. The authors should consider why liquid scintillators have never been used for muon tomography so far. Muon tomography requires high directional resolution, which is why various methods such as emulsion and plastic scintillator cited by the authors have been developed.

The directional resolution and density-change resolution examined by this manuscript using LS are completely insufficient, and the reviewers cannot imagine a measurement target to which this method can be applied. In proposing a new analytical method, you should show some merits of the method (such as a lower detection limit, higher resolution, lower cost, etc. than existing methods). 

Based on the above description, the reviewers have concluded that the manuscript is not suitable for "Applied Sciences".

#1

L33

As pointed out by the authors, the detection efficiency of LS becomes high with increasing volume. However, most important feature of muon tomography is angular resolution. The authors should discuss the influence on angular resolution by changing detector volume.

#2

L38

The authors point out that one of advantage of LS is direction-independent detection efficiency, but in general, muon tomography has a clear target to observe, and reviewers do not see this as an advantage.

#3

L148

Muon survival probability in this study is only 10^-8. This too low to evaluate the performance of this technique. The authors should move the detector to perform higher efficient analysis.

#4

L225

The authors concluded that the results shown in Figure 5(b) match the shape of the Jinping mountain, but the reviewer is unable to find such information in this figure. Additional description should be written both for main text and this figure.

#6

Figure 7

It is written that the sensitivity is low in the region of cosθzen < 0.2, but it looks this region is not shown in the figure.

#7

The authors do not discuss the directional resolution of the method using LS. The directional resolution may be changed depending on the arrangement of the photomultiplier tube, the time resolution of the data taking system, and the volume of the detector. Please discuss about this point. (Related to comment #1.)

#8

The authors conclude that density changes of 50% (or 25 %) can be identified by this method. The reviewers think this is not sufficient and there is no target for application of this method. What is the measurement target of the application aimed at in this research using this method?

Author Response

#-1 On the other hand, the reviewers judged that this manuscript lacked scientific significance. The authors should consider why liquid scintillators have never been used for muon tomography so far. Muon tomography requires high directional resolution, which is why various methods such as emulsion and plastic scintillator cited by the authors have been developed.

We apologize we didn’t emphasize this article aims to the application of Muon Geotomography, imaging large-scale objects. To clarify this point, we modify the sentence of  Line 27 by:”This article focuses on Muon Geotomography. A pioneering application study [16,17] has demonstrated that it is a promising technique for Geological Prospecting on large-scale objects, such as mountains, to find the internal mineral veins, water layers, air cavities, and others.”

As your comments on the directional resolution, we evaluate the impact.  Just for a reminder, the muon multiple scattering effect after traversing 1000 meters in rock will have the 0.5 degree of angle resolution.  For the case we study, the current resolution 5 degree is sufficient. Definitely, there is a room to improve.

#0 The directional resolution and density-change resolution examined by this manuscript using LS are completely insufficient, and the reviewers cannot imagine a measurement target to which this method can be applied. In proposing a new analytical method, you should show some merits of the method (such as a lower detection limit, higher resolution, lower cost, etc. than existing methods). 

Please see the previous answer. As for the merits, we have explained in the introduction section.

#1 (L33) As pointed out by the authors, the detection efficiency of LS becomes high with increasing volume. However, most important feature of muon tomography is angular resolution. The authors should discuss the influence on angular resolution by changing detector volume.

It is similar to #-1, which has been answered.

#2 (L38) The authors point out that one of advantage of LS is direction-independent detection efficiency, but in general, muon tomography has a clear target to observe, and reviewers do not see this as an advantage.

It is similar to #-1. We apologize again, we are discussing Muon Geotomography.

#3 (L148) Muon survival probability in this study is only 10^-8. This too low to evaluate the performance of this technique. The authors should move the detector to perform higher efficient analysis.

We partially disagree, since a simulation study extended from the experiment under 2400 m overburden is already sufficient for the cases we concern. We agree that we have a better demonstration if we can move the detector to a mountain with overburden less than 1000 m.

#4 (L225) The authors concluded that the results shown in Figure 5(b) match the shape of the Jinping Mountain, but the reviewer is unable to find such information in this figure. Additional description should be written both for main text and this figure.

Agree. We modify Line 226 by:”There are event number excesses in four main directions (two in the south and two in the north) in the radar image, which match the ravines as seen in the satellite image.” , and add this description into the caption of Figure 5.

#6 (Figure 7) It is written that the sensitivity is low in the region of cosθzen < 0.2, but it looks this region is not shown in the figure.

Agree, we modify the captions of Figure 7&8: “The MT is not sensitive to the regions with ${\rm cos}\theta_{\rm zen}<0.2$, due to the low event rate.” to “We do not draw the regions with ${\rm cos}\theta_{\rm zen}<0.2$, where the MT is not sensitive to due to low event rate.”

#7 The authors do not discuss the directional resolution of the method using LS. The directional resolution may be changed depending on the arrangement of the photomultiplier tube, the time resolution of the data taking system, and the volume of the detector. Please discuss about this point. (Related to comment #1.)

See the answer #-1

#8 The authors conclude that density changes of 50% (or 25 %) can be identified by this method. The reviewers think this is not sufficient and there is no target for application of this method. What is the measurement target of the application aimed at in this research using this method?

It has been answered in #0.

Reviewer 5 Report

This is an interesting an original proposal for a new type of detector for muon radiography.

The main obvious concern is about the costs. I understand that the authors had the opportunity to recycle a device built for other purposes, but the practitioners may be concerned that the advantages of this new design do not compensate the extra costs with respect to, for example, hodoscopes built with solid plastic scintillator bars. I suggest the authors to add a paragraph somewhere to comment on that.

I am attaching an annotated version of the manuscript where the author can find several minor comments and suggestions.

Author Response

The main obvious concern is about the costs. I understand that the authors had the opportunity to recycle a device built for other purposes, but the practitioners may be concerned that the advantages of this new design do not compensate the extra costs with respect to, for example, hodoscopes built with solid plastic scintillator bars. I suggest the authors to add a paragraph somewhere to comment on that.

The detector is designed for neutrino experiments, and the cost is $200,000. For muon imaging, the cost can be reduced to half or less. The current detector is not a commercial detector, so the cost is not given in the article. Readers can estimate the cost according to the market price through the description in Section 2.2.1.

Article Title:

We decide to keep the original title for brevity.

Figure 1:

Thanks, we change the label of x axis, and modify the title of x axis: “Log10(Energy/GeV)” by “Energy [GeV]”.

Line 103:

This sentence is indeed misleading, and it is only a supplement to the previous sentence, so we delete it.

Line 176-180:

We delete the misleading and overly detailed description: “Therefore…$\left({\rm cos}\alpha_n, \beta_n \right)$.”.

And we modify “We require the entry point to be uniformly distributed on the detector hemisphere facing the muon direction Pn.” by “The entry point is sampled uniformly from the detector hemisphere facing the muon direction P_n.”.

Line 239:

Thanks, “density variation regions’ parameters” is modified by “parameters of the density variation regions”.

And other comments and suggestions are fixed.

Round 2

Reviewer 4 Report

All comments made by the reviewer have been answered in the revised manuscript.